# Experimental Investigation on the Vertical Ductility of Rectangular CFST Columns Loaded Axially

**DOI:** 10.3390/ma15062231

**Published:** 2022-03-17

**Authors:** Bartosz Grzeszykowski, Elżbieta Danuta Szmigiera

**Affiliations:** Faculty of Civil Engineering, Warsaw University of Technology, Al. Armii Ludowej 16, 00-637 Warsaw, Poland

**Keywords:** ductility, pancake type collapse, composite columns, CFST, steel, concrete, experimental research, axial load

## Abstract

A total of 5 steel and 21 rectangular composite concrete-filled steel tube (CFST) columns of moderate slenderness were tested to investigate their ductility under axial compression. The importance of the vertical ductility of columns was discussed, and a novel ductility measure was proposed and utilized to examine the ductility of tested specimens. The analyses showed that the ductility of axially compressed CFST columns highly depends on their failure mode. The key feature influencing the ductility is their ability to dissipate the energy of imposed loads. The larger the volume of a material that may permanently deform and consequently dissipate the energy, the greater this ability. In consequence, the ductility of specimens exhibiting local failure mode was higher in comparison to the columns that underwent global or mixed global—local failure. It was found that both steel and composite columns were able to carry axial loads in the post-critical state; but due to the limitations of local buckling of the steel cross-section in the concrete core and concrete confinement, all tested composite columns showed greater ductility than their steel counterparts.

## 1. Introduction

The vertical ductility of columns is an essential parameter when considering the responses of structures subjected to accidental events, such as fires, impacts, strong earthquakes, and explosions (Figure 1a–c). Such triggering events can, in particular, lead to scenarios involving collapse. Such scenarios are depicted in Figure 1d,e, in which columns of the entire floor of a high-rise building collapse, causing the higher floors to impact the supporting columns of the lower part of the building. In the first case (Figure 1d), the axial bearing capacity or the vertical ductility of columns is high enough to transfer the temporary impact loads to the foundations and the structure retains its overall stability. In the latter case (Figure 1e), the opposite is the issue and a cascade failure occurs, leading to a progressive collapse of the entire building called a “pancake” type collapse. If the dynamic impact force is lower than the axial bearing capacity of the columns, the collapse does not occur. Otherwise, the collapse may be arrested only if the available vertical ductility of columns is greater than the ductility demand.

In the literature, the word ductility is often used to describe the deformability of tested specimens and to categorize them in terms of their ductility without using any direct measure [2,3,4,5,6,7,8,9]. In such cases, the rate of change of the force in the post-peak part of the equilibrium path is analyzed in a qualitative manner. The ductility of structural members can also be directly calculated using various measures, usually referred to in the literature as ductility indexes. Most often, ductility values are defined as the quotients of generalized displacements, such as deflections, rotations, or curvatures [10]. Two main types of analyses including the vertical ductility of columns can be distinguished:Structural design analysis;Comparison of the ductility of structural elements.

In the first case, the ductility of the structure is calculated in order to check its stability in an accidental event. The calculations check whether the ductility demand for a given excitation is lower than the available ductility of a structure. In the latter, the ductility is calculated as one of many parameters in addition to strength, stiffness, durability, etc., which characterize the elements tested and enable a comparative analysis of a given design solution in relation to other types of structures. Depending on the measure of ductility adopted for calculations, the ductility of elements may differ [11].

Several definitions of vertical ductility measures of columns have been used in the literature (see Figure 2). All of them rely on the characteristic points of the equilibrium paths, such as *x_y_*, *x_m_*, and *x*_0,85_, which denote yield displacement, the displacement at the ultimate strength, and the post-peak displacement when the load carrying capacity has undergone a 15% reduction in load, respectively. The available ductility of a column is in fact the post peak displacement for a given reduction in load, and *x_y_* or *x_m_* are used mainly for normalization purposes.

The ductility measures defined in Figure 2 are analogous to horizontal ductility measures used in seismic analysis for the reduction of elastic horizontal seismic forces [29]. However, in the case of vertical ductility of columns, they do not have a clear physical interpretation. Relating the post peak vertical displacement of a column to the displacement at its first yield is not essential to verify the stability of columns subjected to accidental axial loads. Important, however, is the relationship of column load capacity and its external axial force in the element prior to the extreme event (column utilization ratio) in relation to its ductility.

In this paper, the importance of the vertical ductility of columns is discussed in the context of their stability under accidental loads. Based on the concept of the ultimate bearing capacity of columns for extreme situations, a novel vertical ductility measure of axially compressed columns is proposed. A series of experiments was carried out on axially compressed slender (*L*/*D* = 15, where for circular cross-section *D* is its diameter and for rectangular one *D* is a length of its shorter side, *L*—column length) and moderately slender (*L*/*D* = 9.4) steel and composite columns. In total, 21 composite and 5 steel elements were tested. A comparative analysis of the vertical ductility of the tested specimens was conducted. The variable parameters were structural solution—steel or composite concrete-filled steel tubes (CFST)—column slenderness, wall thickness, concrete strength, and grade of structural steel. The experimentally obtained equilibrium paths and failure modes were analyzed for comparative purposes.

## 2. Ductility of CFST Columns—State-Of-The-Art

Figure 3a presents the equilibrium paths of steel, concrete, and CFST stub columns. It can be seen that the ductility of the concrete-filled steel tube is significantly enhanced when compared to those of the steel tube and the concrete alone. The combination of two materials in CFST columns, steel and concrete, takes advantage of the pros and eliminates the cons of both of them. The main factors influencing the increase in ductility of composite stub columns are the concrete core counteracting the local buckling of the walls of the steel section towards the inside and the confinement of the concrete core by the steel jacket. The local buckling of the steel wall towards the center causes the buckled part not to carry load in the post-peak part of the equilibrium path, which causes a significant decrease in the ductility of the entire element [30]. The increase in ductility of concrete caused by its confinement results in overall improvement of the ductility of the CFST stub column [31,32].

In [33,34], the load capacities of 12 circular and 7 rectangular CFST columns were analyzed. It was found that for columns with ratios *L*/*D* < 11, their load capacities were greater than expected in calculations. In more slender elements, the global buckling determined the load bearing capacities of the elements. The reasons were tied to the triaxial stress state in the concrete resulting from its confinement by the steel jacket, which caused an increase in the load-bearing capacities of the elements. After applying sufficient longitudinal strain to concrete sufficient for microcracks to appear, the transverse dimensions of the concrete core increase, resulting in an additional confining pressure. Significant increases in the load capacity are recorded only in circular sections in which the steel jacket provides a uniform transverse pressure. This limits the deformations of the concrete and thereby ensures a significant degree of constraint, increasing both the load-bearing capacity and the ductility. The efficiency of concrete confinement in rectangular CFST cross-sections is less than that in circular ones [35]. In rectangular CFST columns, the greatest confinement of concrete occurs at the corners of the profile and in the middle of the cross-section in the area defined by the four curves shown in Figure 3b [31].

**Figure 3 materials-15-02231-f003:**
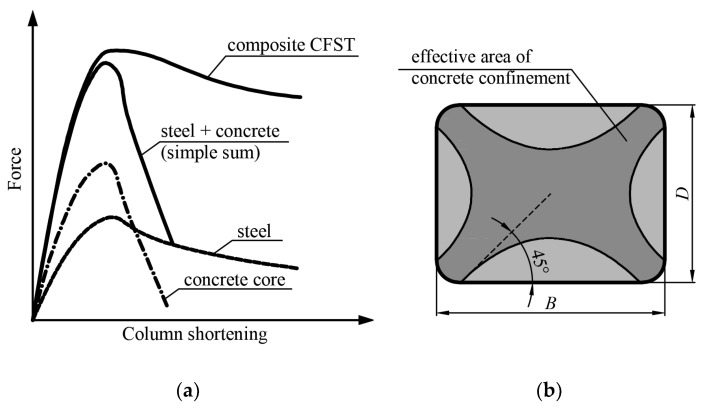
(**a**) Axial compressive behavior of CFST steel, concrete and CFST stub columns, based on [36]; (**b**) effectively confined concrete in a rectangular column, based on [31].

In [37], test results of almost 270 CFST columns with circular, octagonal, and rectangular cross-sections with 2 ≤ *L*/*D* ≤ 9 and 19 ≤ *D*/*t* ≤ 75 were presented. As a result of the analyses, three types of behavior of stub CFST columns were distinguished: 1—with hardening, 2—nearly perfectly plastic behavior, and 3—with softening. All columns with circular cross-sections and most of those with octagonal cross-sections, due to the confinement of concrete, were classified into categories 1 and 2. Some octagonal and all rectangular columns were classified as type 3. In [13], the vertical equilibrium paths of 14 CFST circular and 11 rectangular stubs with 4 ≤ *L*/*D* ≤ 4.8 and 17 ≤ *D*/*t* ≤ 47 were analyzed. It was found that the ductility of circular columns, defined as *μ* = *x_m_/x_y_* (see Figure 2), is greater than the ductility of columns with rectangular cross-sections. For circular stubs, the ductility is *μ* ≥ 10, and for rectangular, 2 < *μ* < 8. All tested circular columns exhibited type 1 behavior (even with a relatively thin steel jacket), whereas columns with rectangular cross-sections showed the same behavior only with thick steel walls (*D*/*t* < 20). In [13], it was also found that for circular cross-sections the confinement of concrete usually becomes significant only after reaching approximately 90% of the load capacities of tested elements, and the local buckling of steel profiles is equivalent to reaching their ultimate load capacities. In [6] the behavior of circular and rectangular CFST stub columns was analyzed. It was found that an increase in steel ratio can change the failure of concrete from being localized into more or less uniform. This change corresponds to the transition of the axial response of the specimen from softening into a ductile response; that is, the axial ductility of the member is increased with by an increase in the steel contribution ratio: *ξ* = *A_a_f_y_*/*A_c_f_c_*, where *A_s_* and *A_c_* are the areas of the steel and concrete cross-sections, respectively, and *f_y_* and *f_c_* are the yield strength of the steel and concrete, respectively. The use of a thicker steel casing caused greater confinement of the concrete, which resulted in crushing of the concrete over a greater length of the stub column, which increased its ductility.

The use of a higher class of concrete increases the load capacities of CFST stub columns but decreases their ductility. In [7], it was found that the use of high-strength concrete (*f_c_* = 173 ÷ 184 MPa) in circular CFST columns resulted in a significant decrease in the vertical ductility of the tested elements. This phenomenon was attributed to the inadequate ability of steel casing (*f_y_* = 420 MPa) to confine high strength concrete. It was stated that in order to increase the efficiency of confinement, thicker steel walls should be used and the steel contribution ratio in the cross-section should be at least 30%. Similar conclusions were drawn from the analyses carried out in [37], where it was additionally recommended that the maximum value be *D*/*t* = 30.

In [38] the ductility of rectangular CFST stub columns was analyzed. Figure 4a shows the ductility index as a function of the steel contribution ratio, and Figure 4b shows DI depending on the aspect ratio of the rectangular section *β* = *h*/*b*. Ductility index was defined based on strains DI = *ε*_85%_/*ε_u_*, where *ε_u_* is the strain at the ultimate load, and *ε*_85%_ is the strain when the load falls to 85% of the ultimate load. The increase in the steel contribution ratio had a positive effect on the ductility of rectangular CFST stub columns. The inverse relationship can be noticed when analyzing the influence of the cross-sectional shape factor *β*. Columns with square cross-sections exhibited the greatest ductility, and the elongation of one of the cross-section sided has a negative effect on its ductility. Similar observations were presented in [39].

The influence of the aspect ratio of the cross-section on the ductility of a CFST stub column was also noticed in [20], where elliptical cross-sections were analyzed. As a result of the parametric analysis, it was found that the ductility of the columns decreases with the increase in the eccentricity of the ellipse. It is worth noting that the standard [40] states that the height-to-width ratio of the composite cross-section should be between 0.2 and 5.0, which excludes the use of very elongated sections. In [20] it was confirmed that the increase in the concrete strength class resulted in a decrease in the ductility of the test elements. 

Figure 5 presents typical local failure modes of axially compressed stub columns: steel, concrete, and CFST. The steel element (Figure 5a) undergoes local inward and outward buckling of the wall, whereas in CFST stub columns that exhibit symmetrical elephant foot failure (Figure 5c) only the outward buckling of the steel occurs, along with the crushing of concrete. In CFST stub columns, non-symmetrical shear and sliding failure (Figure 5d) may also occur similar to the typical failure mode of the plain concrete stub (Figure 5b). The difference is that the steel casing prevents the brittle shear failure usual for concrete stub columns and allows solid wedge movement along a shear frictional surface, increasing the ductility of CFST elements.

The above considerations concern mainly the CFST stub columns—eliminating slenderness from the analysis is helpful to identify the behavior of steel and concrete and the mutual interaction between the two materials at the cross-sectional level. In [42] it was found that slender CFST columns with relative slenderness levels greater than $\bar{\lambda }$ = 0.2 exhibited insignificant confinement of their concrete cores by their steel jackets. However, such elements are rarely used in practice, since this type of column is designed to reduce cross-sectional dimensions. Due to global imperfections in the more slender CFST columns often used in practice, the local failure modes presented in Figure 5 are not likely to occur.

Figure 6a,b shows the typical failure modes of very slender steel and composite CFST circular columns, as presented in [42]. In both cases, the ultimate bearing capacity of the columns is related to the global buckling. However, in case of the steel column, in the center of the column the additional local buckling of the more compressed wall towards the inside of the cross-section occurs. In very slender CFST columns (Figure 6b), this phenomenon does not occur because the concrete core prevents the inward deformation of steel. Similar conclusions were drawn in the study of slender elliptical columns [43].

In [27] the behavior of slender and moderately slender RCFST (round-ended concrete-filled steel tube) columns was investigated. The shortest sides of a rectangular section were converted into half circles, creating ovular, rounded ends. It was found that in the case of moderately slender columns, failure occurred as combination of global and local failure, as shown in (Figure 6b). The more compressed side of the cross-section underwent local buckling, but the concrete filling prevented its deformation towards the inside. For very slender elements, the local buckling did not occur, and the failure was dictated by an elastic flexural global buckling, similar to that shown in Figure 6c. The ductility analysis showed that with the increases in *B*/*t* and *B*/*D*, where B and D are the larger and smaller cross-section dimensions, respectively, the ductility of the elements decreased. The decrease in ductility also occurred with an increase in the class of concrete used.

In the literature, there is lack of investigations considering the ductility of rectangular CFST columns with moderate slenderness that are most used in practice. Either stub or very slender CFST columns were tested, and their behavior is described in the literature. Therefore, it was necessary to carry out the investigation on the vertical ductility of axially compressed, moderately slender, rectangular CFST columns.

## 3. Proposal for the Vertical Ductility Assessment of Columns

Figure 7 shows two equilibrium paths of columns with the same ultimate bearing capacity that exhibit sufficient (Figure 7a) and insufficient (Figure 7b) ductility under dynamic axial loads. The same scale of axes was kept for Figure 7a,b; therefore, it can be seen that the ductility of the column whose equilibrium path is depicted in Figure 7a is greater in relation to Figure 7b. If the element is sufficiently ductile, after the imposed loads are withdrawn, there is a new stable equilibrium point (marked as X in Figure 7a). In this case the column maintains its residual load bearing capacity and does not lose its stability after the extreme event. On the other hand, if the ductility of the column is insufficient (Figure 7b), then after the accidental loading the next point of equilibrium does not exist and the column collapses. On analyzing the data in Figure 7, it can be concluded that the preloading force prior to the imposed loading is not less important. The smaller the utilization ratio of the column, the lower the vertical ductility demand because a new equilibrium in the post-peak may occur for a column with less residual strength.

In Figure 7, the ultimate axial bearing capacity of the column is denoted as *N_R_*, whereas *N*_0_ = *ωN_R_* is an equivalent axial force in the column prior to the extreme event. A reduction parameter *ω* is used to account for the influences of various combinations of actions’ coefficients and material safety factors in the permanent and accidental design situation, the reserve factor, and the strain rate behavior of the steel and concrete. Assuming that the utilization of the element in the permanent design situation is equal to 100% and that the typical safety factors for concrete and steel are equal to 1.4 and 1.0, respectively, an upper limit of this coefficient for composite steel-concrete columns is equal to approximately 0.8. Increasing the column’s reserve factor in permanent design situation results in a decrease in the *ω* coefficient.

An energetic measure of the ductility *μ_ω_* of axially compressed columns based on the displacements of the column top in the pre- and post-peak periods, *x*_0_ and *x_ω_*, respectively, is proposed (see Figure 8). The values of these displacements depend on the shape of the equilibrium path and on the value of the *ω* coefficient. The ductility measures based only on the quotients of displacements do not consider the full shape of the equilibrium path, but only the chosen points. The proposed energy measure has the advantage of considering the entire shape of the equilibrium path, both in the pre- and post-peak up to the displacement *x_ω_*. *E*_0_ is the energy (mostly elastic) accumulated in the column at *N*_0_ load, and *E_ω_* is the energy in the column (accumulated and dissipated) when the force drops to *N*_0_ in the post-peak part of the equilibrium path. In fact, the energy *E_ω_* is the physical measure of the vertical ductility of a given column, and the energy *E*_0_ is introduced for normalization purposes. The advantage of the proposed measure of ductility is also the fact that for a given equilibrium path, *E*_0_ and *E_ω_* can be easily determined, since both *x*_0_ and *x_ω_* are explicitly defined as points of intersection of the equilibrium path with the ordinate *N*_0_. This avoids imprecise definition of the yield and failure points used in the conventional vertical ductility measures adopted from horizontal ductility factors widely used in seismic analysis [29]. Those have a physical interpretation for horizontal ductility, but not necessarily for the vertical one. The failure point in the proposed vertical measure is defined as the point on the equilibrium path for which for a given force *N*_0_, failure of the column will occur.

The number of publications on modelling of the behavior of composite CFST columns continues to rapidly rise. Fiber element models [45,46] and FEM simulations [3,6,9,11,20,27,47,48,49,50,51] are proven to be capable of accurately predicting the shapes of equilibrium paths of composite CFST columns, including their post-peak behavior. Recently, artificial neural network (ANN) models [52] and machine learning algorithms [53] were employed to accurately predict the ultimate strength of CFST columns. Further improved ANN-based models were then used to predict and plot the complete axial load-shortening curves of concentrically loaded rectangular and circular CFST columns that reasonably matched the experimental results [54]. This approach will hopefully allow determining the vertical ductility of CFST columns without performing costly and time consuming experimental and FEM analyses.

## 4. Experimental Research—Methodology

The experimental research was conducted on 26 cold-formed steel rectangular hollow section columns made of S235JR and S355JR-grade steel. The length of each element was 750 mm. The analysis included 3 types of composite steel–concrete column. For comparative studies, steel columns that had not been filled with concrete were also tested. A, B, and D describe columns made of RHS100 × 50 × 5, RHS100 × 50 × 3, and RHS120 × 80 × 3 profiles, respectively. S1 and S2 indicate the steel grades S235 and S355, respectively, and C1 and C2 indicate the concrete classes C20/25 and C50/60, respectively. In the experimental analysis, the variable parameters were the type of column (steel, composite), slenderness, wall thickness, concrete strength, and steel grade. Standardized steel and concrete samples were tested to obtain the material properties. A detailed description of the material and geometrical properties is provided in Table 1 and depicted in Figure 9. Table 1 columns contain, respectively, the notation of the specimen; steel cross-section; steel grade; yield strength of steel; concrete strength; quotients *D*/*t* and *L*/*D*, where *D*—height of the steel profile, *t*—wall thickness, *L*—total length of the column; $\bar{\lambda }$—relative slenderness; *ξ*—steel contribution ratio; the number of elements tested in each series. The columns from series D were of moderate slenderness, and elements from series A and B were considered slender in relation to columns from series D.

The columns were compressed in a hydraulic press. The applied force vs. column shortening was measured, along with the vertical strains of the steel profiles using strain gauges placed mid-length. In addition, the digital image correlation method (DIC) [45] was used to measure the displacement in composite members. A random in position but uniform in size speckle pattern was applied to the surface of each CFST sample (see Figure 10a). In order to maximize the contrast of images white primer and black car paint were used. In order to ensure proper adhesion of the paint, the samples were sandblasted prior to the application of the coating. The displacements were measured using two cameras taking pictures of a certain rectangular area of each tested composite column, and the coordinates of selected points located in the measuring area were determined. The measured coordinates, which were related to the coordinates of the same points measured in the first image of the unloaded sample, were then used to calculate the components of the displacement and strain field. The calibration was performed in accordance with [55] using a calibration panel with two scale bars of MV 175 × 140. The calibration deviation was equal to 0.019 pixels which corresponds to an accuracy of measurements equal to approximately 5 μm.

In order to obtain equilibrium paths in the post-peak parts of the equilibrium paths of columns, a displacement control method was used in the analysis. For technical reasons, due to expected exceptionally large deformations of investigated elements and therefore a possibility of uncontrolled ejection of columns from the hydraulic press after buckling, the knife edged boundary conditions simulating pinned-pinned support boundary conditions were not used. The test elements were each placed on a rigid, non-deformable steel plate and compressed by a spherical traverse, embedded in a spherical seat; hence, columns’ ends were partially fixed due to friction. The experimental setup was as shown in Figure 10b.

## 5. Results of the Experimental Research and Discussion

### 5.1. Failure Modes

The typical failure modes of the analyzed columns are shown in Figure 11. Three types were observed: a combination of global and local buckling (from this point called only the global buckling), local shear and sliding, and elephant foot (splitting) failure. Global buckling occurred in all slender columns from series A (Figure 11a) and in one specimen, BS1C2_1, from series B (Figure 11b). In all remaining slender columns from series B, the failure occurred due to the local shear and sliding of the concrete core (Figure 11c). In only one moderately slender specimen, DS2C1_1, was elephant foot (splitting) failure observed (Figure 11d). The failure of all remaining specimens from series D occurred due to shear and sliding of inner concrete (Figure 11e).

In moderately slender columns from series D, all failure modes were of the local type, and in slender columns with 5 mm wall thickness (series A), global type. This was due to the relative slenderness of the tested elements, which in the case of columns from series D was on average $\bar{\lambda }$ = 0.33, and in the case of columns from series A, $\bar{\lambda }$ = 0.57. It is worth noting that in the case of columns from series B with similar slenderness to elements from series A ($\bar{\lambda }$ = 0.53), but with thinner walls (*t* = 3 mm), the failure mode of most test elements is local. Only in one specimen BS1C2_1 is the failure mode global.

The steel contribution ratio *ξ* is used in the literature to estimate the effectiveness of concrete confinement by the steel jacket in a composite column [38,49]. The specimens from series D, in which mostly the local shear and sliding failure occurred, exhibited the smallest values of this coefficient, 0.90 ≤ *ξ* ≤ 1.51, (see Table 1). The symmetric “elephant foot” failure type occurred only in specimen DS2C1_1, which was characterized by the highest value of steel contribution ratio among all elements from series D (*ξ* = 1.51). This suggests the influence of concrete confinement on the types of local failure modes of moderately slender composite columns. The greater this phenomenon, the greater the probability of a symmetrical failure mode such as “elephant’s foot.” On the other hand, in another column of this series, DS2C1_2, the “elephant’s foot” failure did not occur, which suggests the high complexity of phenomena occurring in rectangular CFST columns and their random nature.

In slender elements from series A (*t* = 5 mm) in which the steel contribution ratio had of the highest values, 3.18 ≤ *ξ* ≤ 4.30, only global buckling was observed. As a result of significant confinement of concrete by the steel jacket, the concrete core was neither crushed (“elephant foot” failure) nor sheared (“shear and slip” failure). In these elements, the local load-bearing capacity and ductility of concrete increased by its confinement are high enough that they do not determine the load-bearing capacity of the entire element. Therefore, global sway buckling occurs, and in the middle of the column a plastic hinge is formed. Only after the buckling of the entire element, in the compression zone of the middle section, does the concrete undergo one-sided crushing, and the compressed part of the steel jacket is subject to local buckling (see Figure 11a).

In slender elements from series B (*t* = 3 mm), the confinement of concrete (1.52 ≤ *ξ* ≤ 1.97) was not high enough to trigger the global buckling failure mode before the local shear of concrete. This was the case only in one specimen marked BS1C2_1 (see Figure 11b), in which the occurrence of global buckling before local concrete core shear could have been caused by a larger arc imperfection of the element or additional imperfections caused by the gusset plates welded not ideally perpendicular to the axis of the column.

### 5.2. Qualitative Analysis of Equilibrium Paths

Experimentally obtained equilibrium paths of the analyzed columns are presented in Figure 12 and Figure 13, in which the vertical axes have been normalized with respect to the ultimate bearing capacity of each column.

Regardless of the slenderness of the columns, the class of concrete, or the strength of steel, in all test series, the rate of change of the force in the post-peak part of the equilibrium paths is fairly consistent in composite columns in relation to their steel counterparts. Indeed, normalized equilibrium paths of composite columns were always above the paths of their steel counterparts in terms of strength, and this difference increased with increasing displacement. This was due to the concrete core counteracting the local buckling of the walls of the steel section towards the inside. In steel columns, local buckling causes the buckled part of the cross-section to effectively stop carrying the loads. Consequently, this reduces the ductility of the entire column. When the concrete core counteracts the deformation of the steel jacket towards the center, the deformability of such a cross-section increases, as the greater part of the steel section is still able to bear loads. Moreover, the steel jacket provides concrete confinement, which increases the ductility of the inner concrete, and therefore, the ductility of the entire element. 

Figure 12e depicts the normalized equilibrium paths of two specimens from series BS1C2 whose failure modes were different. Specimen BS1C2_1 exhibited global buckling failure (see Figure 11b), whereas in specimen BS1C2_2, local shear and sliding failure occurred (see Figure 11c). The implications of this can be seen in Figure 12e. The ductility of specimen BS1C2_2, considering displacements greater than 15 mm, is much larger than the ductility of BS1C2_1. In all columns which exhibited local failure mode, as a result of additional shear caused by friction and jamming of the crushed concrete within the steel jacket, there was a cyclic increase and decrease in force. Elements where this type of deformation occurred exhibited significantly smaller horizontal displacements. Therefore, after reaching the ultimate bearing capacity of the weakest cross-section of a member, in subsequent cross-sections, further deformation may occur, incorporating more material along the length of the column in the dissipation of energy.

A comparison of normalized equilibrium paths of steel columns from all series can be seen in Figure 14a. By analyzing the presented data, it can be concluded that the influence of the steel grade (S235 or S355) on the ductility of steel columns was insignificant for both 3 mm (series D) and 5 mm wall thickness (series A), regardless of their slenderness. It can also be seen that the columns made of thicker plates exhibited significantly greater ductility. Moreover, the equilibrium paths of columns from series A showed greater flattening after reaching their limit load capacities. Before reaching the vertical displacements equal to approximately 8 mm, the presented relationships resemble an elastic-ideally plastic response by the structures. The equilibrium paths, and therefore the ductility of columns from series B and D, are similar.

The influence of the wall thickness of the steel jacket on the behavior of slender composite columns can be seen in Figure 14b, which depicts the normalized equilibrium paths of elements from series AS1C1 and BS1C1. It can be noticed that up to the vertical displacements equal to approximately 15 mm, the equilibrium paths of the columns made from plates 3 mm thick lie below the paths for columns made of thicker steel sheets, and therefore, exhibit smaller ductility. After reaching approximately 10–15 mm vertical displacement, the equilibrium paths of columns from series B start to increase, and therefore, those elements exhibited greater post peak strength (and ductility) for larger displacements in comparison to columns from series A. This was caused by the local failure mode of all elements from series BS1C1 (see Figure 11c).

The influence of the concrete strength on the ductility of moderately slender composite columns can be seen in Figure 15a, where both standard and normalized equilibrium paths of elements from series DS1C1 and DS1C2 are presented. On analyzing the presented data, it can be concluded that the use of stronger concrete increases the ultimate strength of columns, but in the post-peak part of the equilibrium path, the residual bearing capacity was equal to approximately 450 kN and was a similar level for all depicted elements. This resulted in lower ductility of moderately slender CFST columns filled with stronger concrete that exhibited a local type of failure. It can be seen more clearly when analyzing the normalized equilibrium paths presented in Figure 15b. In the post-peak zone, the relative force in a column filled with stronger concrete is, for the same displacement, lower compared to the relative force in a column filled with weaker concrete. In other words, the normalized equilibrium paths of the DS1C1 columns lie above the normalized paths of the DS1C2 columns.

A different conclusion can be drawn when analyzing the data presented in Figure 16a, where normalized equilibrium paths of slender elements from series AS1C1 and AS1C2 are presented. It can be seen that the concrete strength had no influence on their ductility, owing probably to the global buckling failure mode of those elements. In columns from series A, most deformations (see Figure 11a) were focused on the plastic hinge formed in the center of the column as a result of the global sway failure mode. The parts of the column above and below the plastic hinge operate approximately as a rigid body; therefore, not much concrete was used in energy absorption during the loading process, as was the case in all elements from series D, which exhibited local failure modes. The confirmation of the above considerations can be found in [34], where the phenomenon of concrete confinement was clearly observed only in stub and moderately slender composite columns (*L*/*D* < 11).

The influence of the steel strength on the behavior of composite columns can be seen in Figure 16b. It can be seen that the normalized equilibrium paths for columns made of stronger steel lie above the equilibrium paths of columns made of weaker steel. This can be explained by the greater concrete confinement in elements made of S355 steel. The average steel contribution ratios *ξ* in the DS2C1 and DS1C1 columns were equal to (see Table 1), *ξ_,_*_DS2C1_ = 1.51 and *ξ_,_*_DS1C1_ = 1.16, respectively.

### 5.3. Quantitative Analysis of Equilibrium Paths

Figure 17 shows the ductility of analyzed columns calculated using proposed energetic measure *μ_ω_*_,0.7_. It was assumed that equivalent axial force *N*_0_ is equal to 70% of the ultimate bearing capacity of columns (*ω* = 0,7, see Figure 8). In the column chart, the same colors indicate composite and steel columns made of the same steel profile and the same class of structural steel. CFST elements filled with C25/30 concrete (designation C1) are shown as rectangular posts, and those filled with C35/40 concrete (designation C2) as posts with rounded edges.

On analyzing the presented data, it can be seen that most of the conclusions drawn on the basis of the qualitative analysis that included the wider spectrum of columns’ top displacements conducted in the previous chapter of this paper have been confirmed. In each case, filling the steel cross-section with concrete caused a significant increase in the ductility of the tested elements (for series A: 35–60%, series B: 23–25%, and series D: 20–83% increase). In slender columns from series A and B that exhibited a global failure mode, the higher strength concrete filling had no significant influence on the ductility of the elements. However, in moderately slender columns from series D that exhibited local failure modes, the ductility of specimens filled with C1 was 22% higher in relation to those filled with C2. Similarly, it can be observed that the ductility of moderately slender columns from series D made of stronger steel S2 exhibited a 33% increase in ductility in relation to columns made of weaker steel S1. The wall thickness had a significant influence on the ductility of slender CFST columns—the ductility of elements from series AS1C1 with *D*/*t* = 10 was almost 2.5 times higher than the ductility of column BS1C1 with the *D*/*t* = 16.7. When analyzing the ductility of steel columns, it can be concluded that the highest ductility was exhibited by elements made of thicker walls, and the steel strength had no influence on the results whatsoever.

### 5.4. Digital Image Correlation—Results

Contour plots of longitudinal *ε_xx_*, transverse *ε_yy_*, and shear *ε_xy_* strains of selected CFST specimens are presented in Figure 18, Figure 19, Figure 20, Figure 21, Figure 22 and Figure 23. It was assumed that the strains describing the relative elongation are positive. Figure 18, Figure 19 and Figure 20 present longitudinal strains of moderately slender columns from series D. The individual strain contour plots are marked with capital letters which correspond to the points marked on the equilibrium path placed in the lower left corner. Points A and B indicate strains before reaching the ultimate load and when the maximum force was reached, respectively. Points C, D and E mark selected places in the post-peak part of the equilibrium path. On the left side, the experimentally obtained failure modes of specimen are presented. A red rectangle indicates the area where displacements were measured. Strain contour plots were rotated 90 degrees counterclockwise to the horizontal position. White spots indicate the areas where the strains exceed the minimum and maximum values assumed while plotting.

On analyzing Figure 18, it can be seen that at point A along the height of the column, some horizontal stripes appeared, denoting places where strain increased several times in relation to other cross-sections of the column. These are areas where local crushing of concrete occurred, and in consequence the core locally lost its ability to transfer compressive stresses. Therefore, in order to maintain a constant force in the weakened cross-section, the strains and consequently stresses in the steel jacket increased. At point B, a few more stripes appeared, and their distribution along the height of the column became approximately uniform. In the post-peak part of the equilibrium path (points C and D), new stripes did not form. In the third from the right strip, where highest strains were observed from the beginning, further localization of strain occurred. Local outward buckling of steel was observed, denoting the area where shear and sliding failure occurred (points D and E).

Significantly different strain localization patterns can be observed in Figure 19 for column DS2C1_2 made of stronger steel and weaker concrete. Until the ultimate load was reached (point B), the horizontal localizations of strains were not observed. On the other hand, in the post-critical part of the equilibrium path (point C), diagonal stripes appeared. This denotes that in specimens made of weaker steel and filled with stronger concrete (DS1C2—horizontal yield areas), a different concrete failure type occurred compared to those made of stronger steel and filled with weaker concrete (DS2C1—diagonal yield areas).

A combination of the two previously described behaviors can be observed in Figure 20, where strain contour plots of DS1C1_1 column made of weaker steel and filled with weaker concrete are presented. It can be seen that although in the pre-peak part of the equilibrium path no strain localization was observed (similarly to the DS2C1_1 column), when the ultimate load was reached (point B), horizontal yield zones appeared, similarly to the DS1C2_1 column. However, as the column shortening increased at the transition from point B to point C, new diagonal yield zones developed.

It was previously observed that the ductility of moderately slender columns from series DS2C1 made of stronger steel S2 exhibited a 33% increase in ductility in relation to columns DS1C1 made of weaker steel S1. This was due to the greater concrete confinement in such elements. This may indicate that there is a correlation between the intensity of the occurrence of diagonal yield zones and the level of concrete confinement in elements that exhibit local shear and sliding or elephant foot-type failure. In CFST columns, concrete confinement is always accompanied by transverse tensile strains of the steel jacket. This can be observed in Figure 21 and Figure 22, which present contour plots of longitudinal and transverse strains in the post-peak part of the equilibrium paths of the DS1C1_2 and DS2C1_1 columns, respectively. It can be seen that the transverse strains in diagonal yield areas are higher in comparison to strains in surrounding zones. Similar diagonal failure mechanisms were observed in [56], where localization of hoop strains in cylindrical FRP-wrapped stub concrete columns was investigated. Depending on the length of the confined concrete cylinder, a number of fully localized failure mechanisms developed, each one of them then involving solid wedge movements along a shear frictional surface, causing stress concentrations in the FRP wrapping and thus precipitating its premature failure.

Figure 23a,b presents the longitudinal strains in slender columns AS1C1_3 (*ξ* = 4.12) and BS1C2_1 (*ξ* = 1.52), respectively. BS1C2_1 was the only specimen from series B that exhibited global buckling failure mode. The top and bottom strain contour plots were captured close to the ultimate force and in the post peak part of the equilibrium paths, respectively. In both specimens, a combination of global and local outward buckling occurred. In the middle of both columns, plastic hinges denoted by elliptical shapes representing the bulged steel jacket (showed in the lower contour plots) were formed. It seems interesting that similarly to series D columns, which all exhibited local failure modes, in the BS1C2_1 element, after reaching the ultimate force, the diagonal yield pattern also appeared, but was located below the plastic hinge that formed in the middle. The reason for this may be larger imperfections in this specimen that caused the global buckling to occur before the local concrete shear could have been fully developed. An outline of similar diagonal yield patterns was also observed in other specimens from series B, but not in any of the specimens from series A. This may indicate that the geometric and material properties of elements from series BS1C2 delimit the global and local failure modes of axially compressed CFST columns.

## 6. Conclusions

In this study, a total of 21 composite CFST and 5 steel rectangular columns of moderate slenderness were tested to investigate their ductility under axial compression. The importance of the vertical ductility of columns was discussed, and a new ductility measure considering the equivalent preloading prior to the extreme event was proposed and utilized to examine the ductility of tested specimens. The test results were presented and analyzed in detail to provide information about the factors influencing the ductility of axially compressed CFST columns.

The conducted analyses showed that the key feature influencing the ductility of axially compressed columns is their ability to dissipate the energy of imposed loads. The larger the volume of a material in the element that may permanently deform (in the case of steel) or crush (in the case of concrete) and consequently dissipate the energy, the greater this ability. Due to the confinement of a concrete core by a steel jacket, and most importantly, the limitation to local buckling of the steel cross-section by the concrete core, all tested composite columns showed greater ductility than their steel counterparts.

The analysis concluded that the vertical ductility levels of axially compressed CFST columns with rectangular cross-sections highly depend on their failure modes. Therefore, the detailed conclusions were divided into two categories, taking into account the failure modes of the analyzed columns. In general, the ductility of specimens with local failure modes was higher in comparison to columns that exhibited global buckling failure.

The conclusions for elements that underwent global buckling:The concrete core failure consisted of crushing the concrete in the plastic hinge on the side of the cross-section with high compressive stresses. In elements with large *D*/*t* ratios, the additional formation of diagonal yield zones in the other cross-sections of the columns occurred.The higher strength concrete filling had no significant influence on the ductility of slender test elements.The increase in the wall thickness had a positive effect on the ductility of slender specimens.For elements that underwent the local failure mode:In shear and sliding failure mode, the concrete core failure consists of the formation of diagonal or horizontal yield zones along the height of the column, causing stress concentrations in the steel jacket, and in the weakest yield zone, shear solid wedge movements along a shear frictional surface develop. In elephant foot failure mode, the concrete core failure consists of crushing the concrete in the weakest cross-section of the column.Symmetrical elephant foot failure mode yielded higher vertical ductility in comparison to shear and sliding failure modes.The use of higher-grade steel increased the vertical ductility of the tested columns.The use of stronger concrete caused a decrease in the ductility of the test specimens.

## Figures and Tables

**Figure 1 materials-15-02231-f001:**
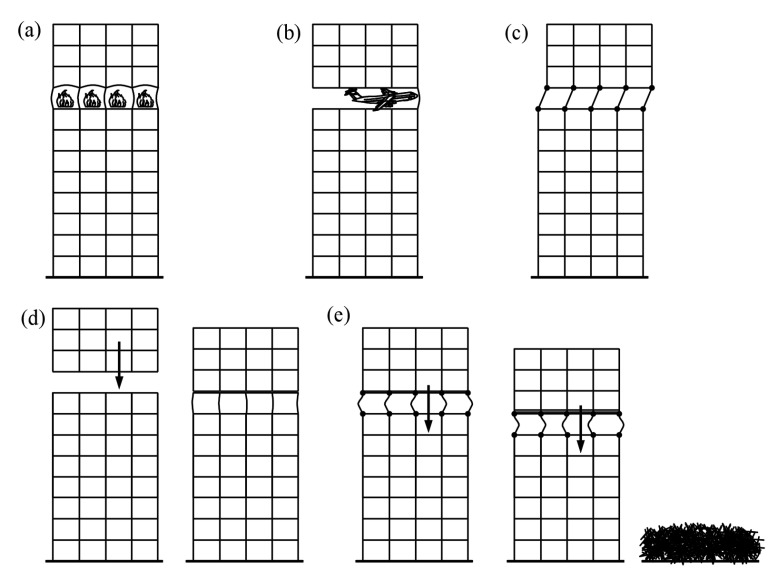
Possible triggering events that can lead to a progressive collapse of a building: (**a**) fire; (**b**) crash with an object or explosion; (**c**) strong earthquake—consequences of the removal of an entire story; (**d**) local failure does not lead to a progressive collapse due to a sufficient strength or ductility; (**e**) progressive collapse, insufficient strength, or ductility of remaining columns, based on [1].

**Figure 2 materials-15-02231-f002:**
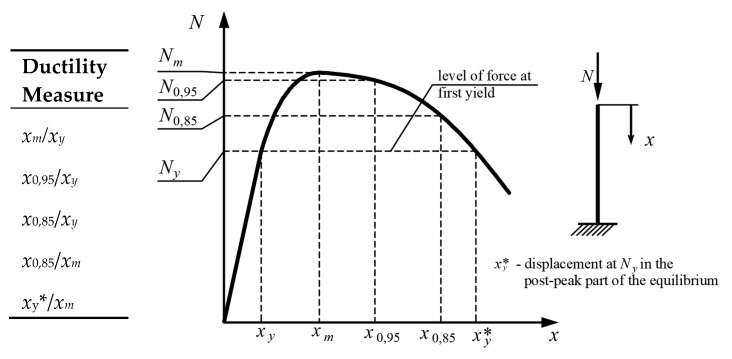
Definitions of vertical displacement ductility factors [12,13,14,15,16,17,18,19,20,21,22,23,24,25,26,27,28].

**Figure 4 materials-15-02231-f004:**
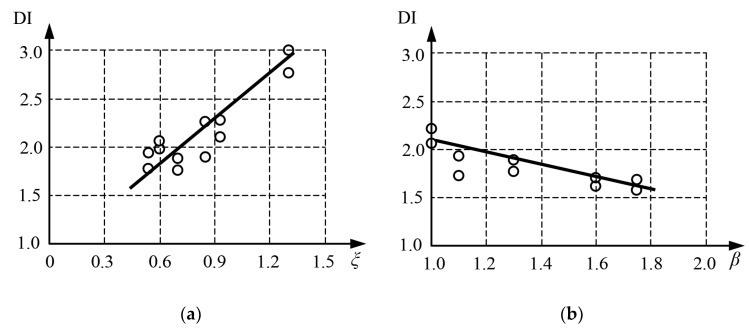
Ductility index DI versus: (**a**) steel contribution ratio *ξ* with β = 1.3; (**b**) the ratio of the cross-sectional dimensions β, with ξ = 0.7, based on [38].

**Figure 5 materials-15-02231-f005:**
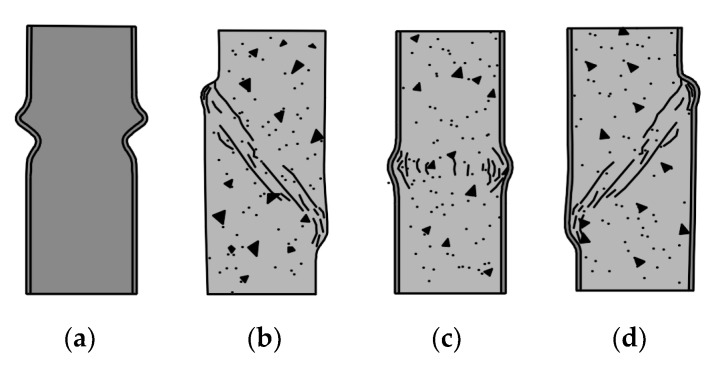
Local failure modes of axially compressed stub columns: (**a**) steel—local buckling, (**b**) concrete without reinforcement—shear (**c**) CFST—elephant foot, (**d**) CFST—shear and sliding, based on [36,41].

**Figure 6 materials-15-02231-f006:**
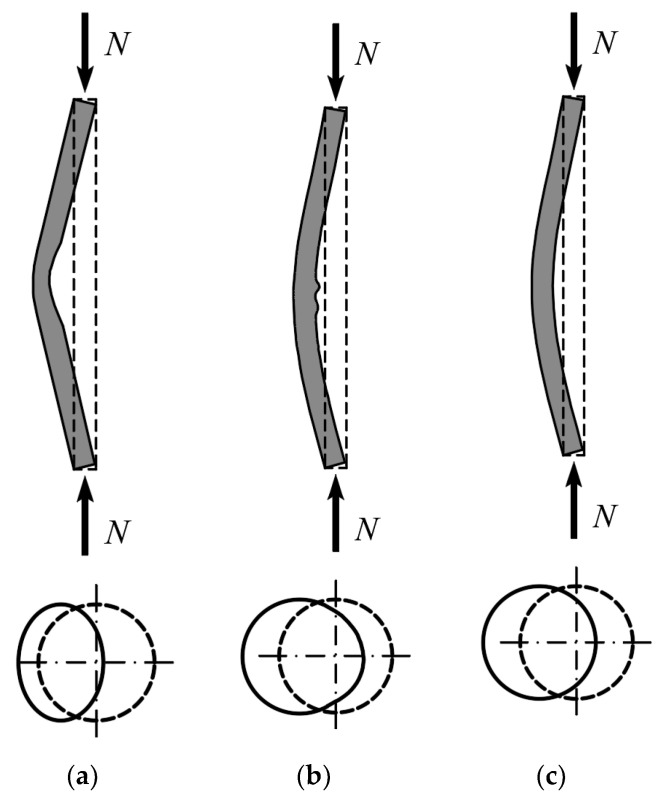
Typical failure modes of slender and moderately slender circular columns: (**a**) steel column, global + local buckling; (**b**) CFST column, global and local buckling; (**c**) CFST column, only elastic global buckling, based on [27,44].

**Figure 7 materials-15-02231-f007:**
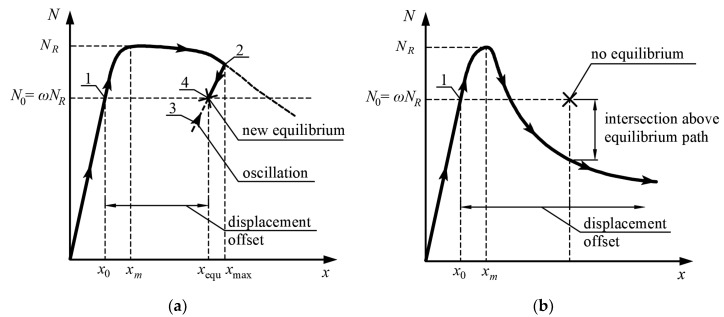
Equilibrium path for a column that exhibits: (**a**) sufficient and (**b**) insufficient ductility. 1—displacement and force in the column just before the excitation; 2—maximum instantaneous displacement; 3—oscillation around the equilibrium position; 4—permanent final displacement.

**Figure 8 materials-15-02231-f008:**
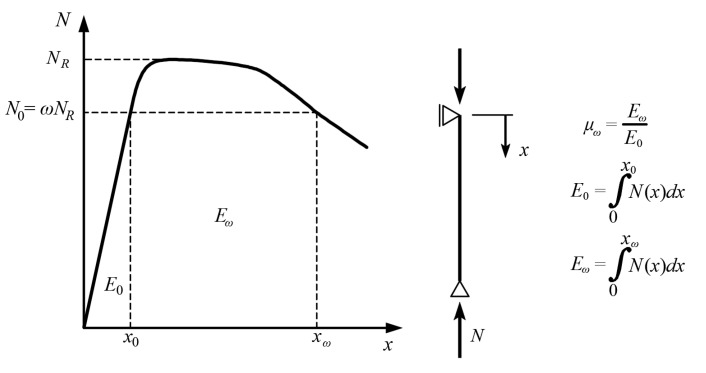
Proposed measure of vertical ductility based on the energy.

**Figure 9 materials-15-02231-f009:**
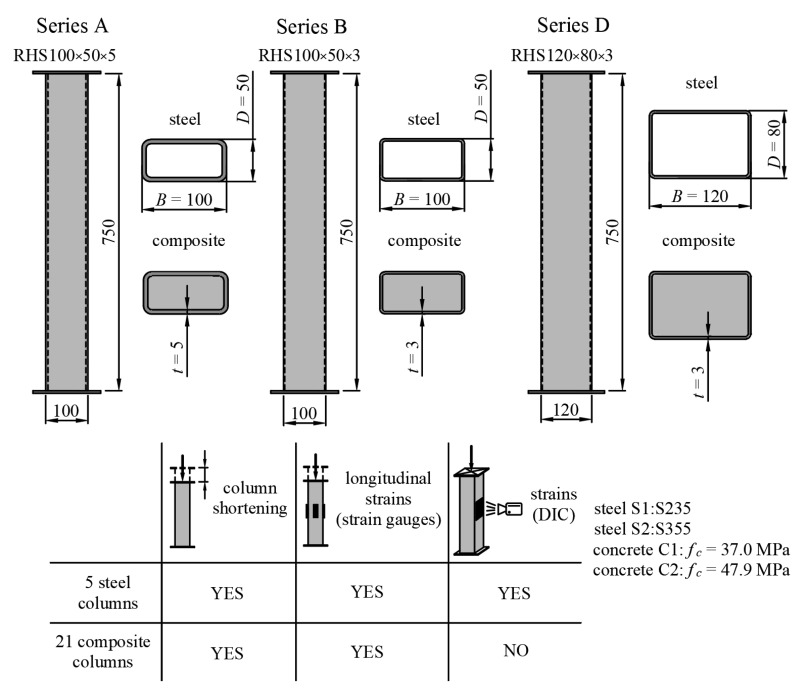
Design schemes and descriptions of the measurements performed on the investigated columns.

**Figure 10 materials-15-02231-f010:**
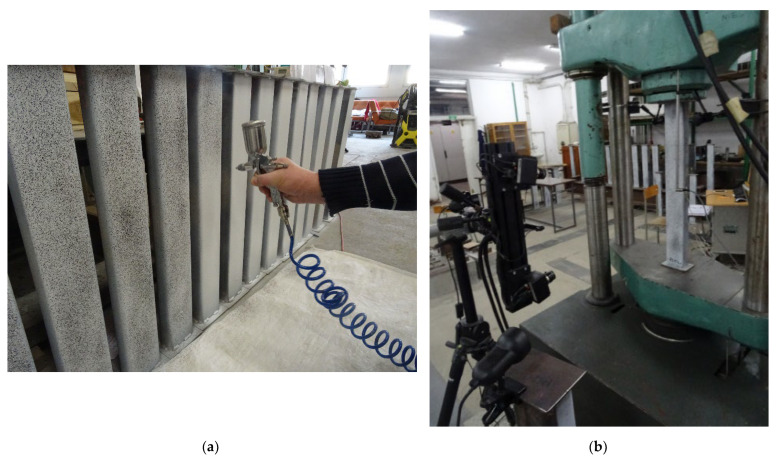
(**a**) Speckle pattern application, (**b**) experimental setup.

**Figure 11 materials-15-02231-f011:**
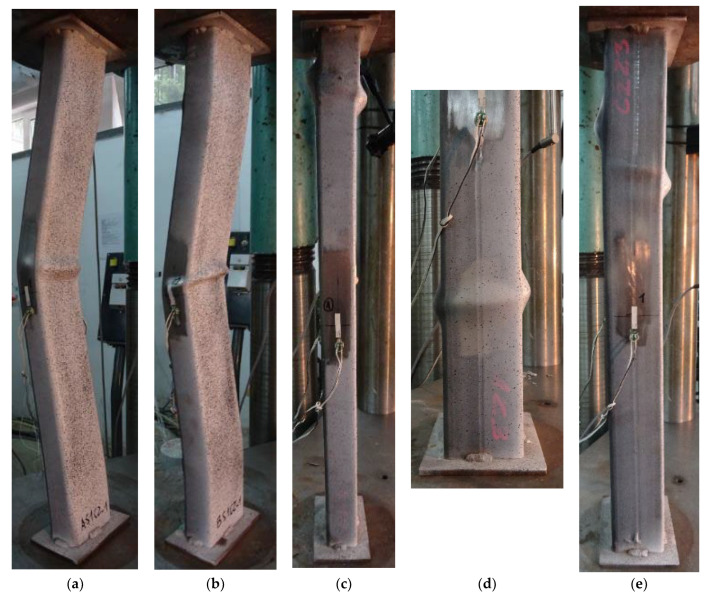
Typical failure modes of composite columns observed during the experiment: (**a**) global buckling of slender columns from series A; (**b**) global buckling of single specimen BS1C2_1; (**c**) local shear and sliding failure of slender columns from series B (except BS1C2_1); (**d**) elephant foot (splitting) failure of single specimen DS2C1_1; (**e**) local shear and sliding failure of moderate slender columns from series D (except DS2C1_1).

**Figure 12 materials-15-02231-f012:**
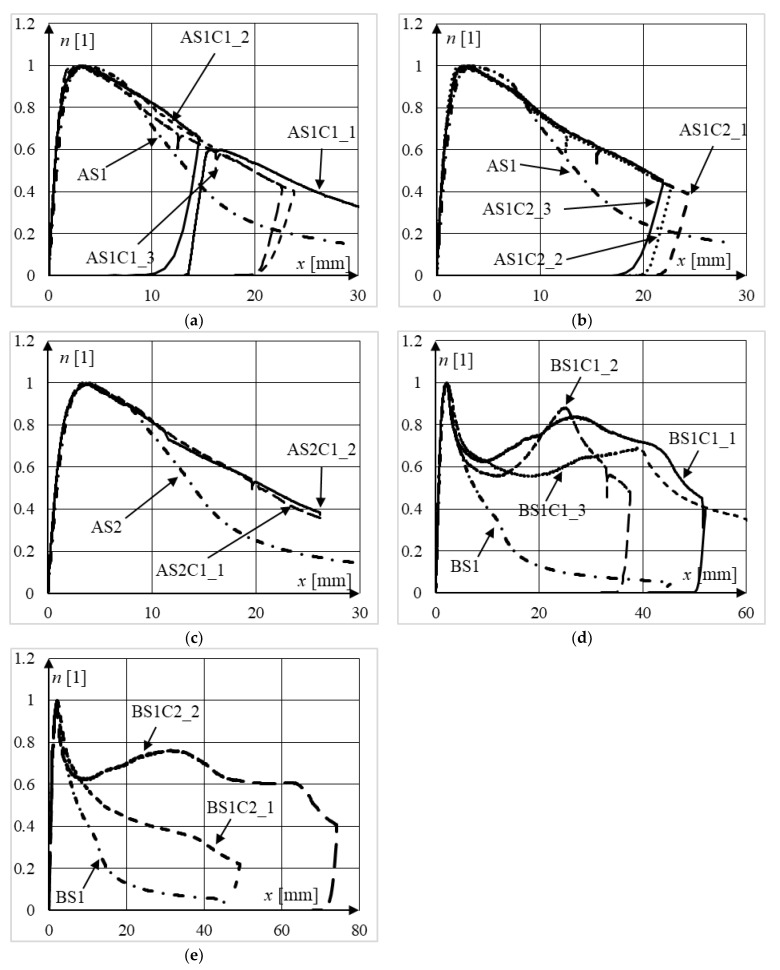
Normalized equilibrium paths of slender elements from series: (**a**) AS1C1; (**b**) AS1C2; (**c**) AS2C1; (**d**) BS1C1; (**e**) BS1C2.

**Figure 13 materials-15-02231-f013:**
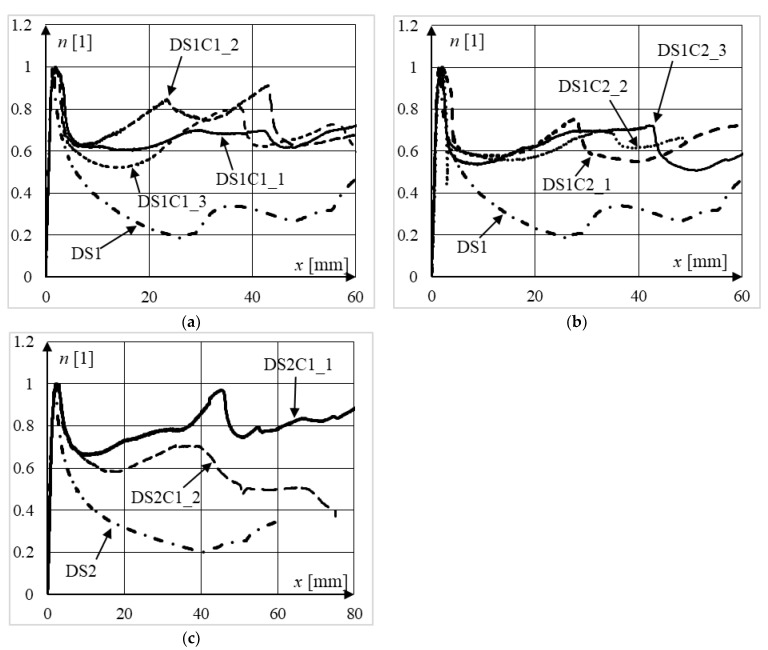
Normalized equilibrium paths of moderately slender elements from series: (**a**) DS1C1, (**b**) DS1C2, (**c**) DS2C1.

**Figure 14 materials-15-02231-f014:**
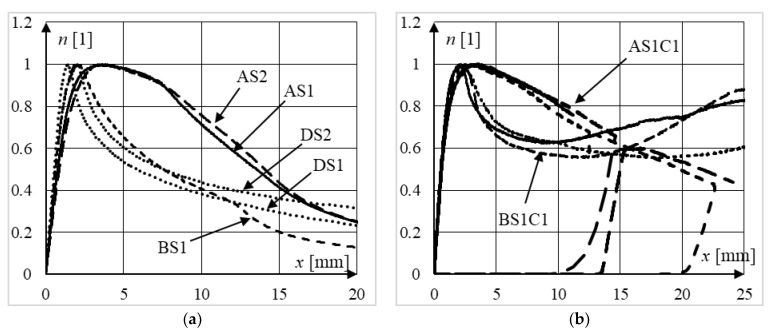
Comparison of normalized equilibrium paths of: (**a**) steel columns from all series; (**b**) slender composite columns from series AS1C1 and BS1C1.

**Figure 15 materials-15-02231-f015:**
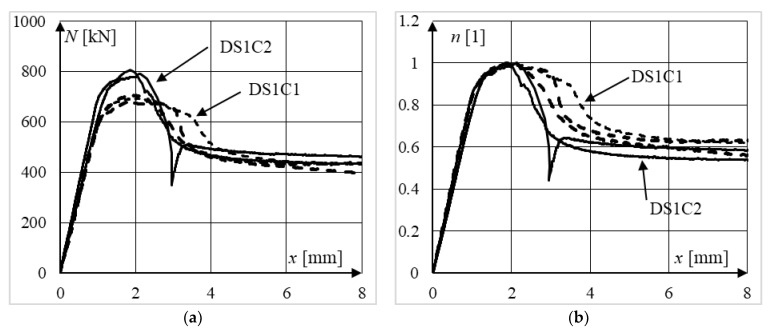
Comparison of (**a**) non-normalized and (**b**) normalized equilibrium paths of moderately slender elements from series DS1C1 and DS1C2 (different concrete strength).

**Figure 16 materials-15-02231-f016:**
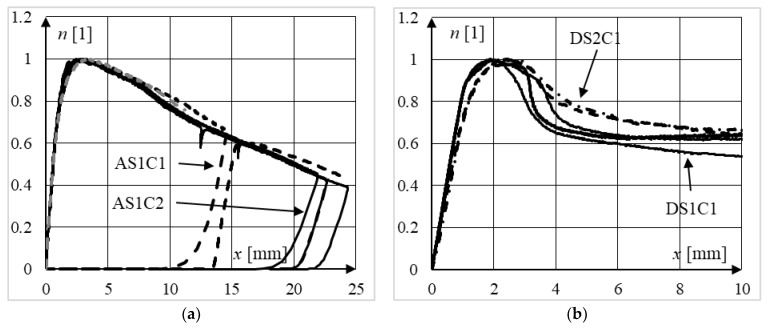
Comparison of normalized equilibrium paths of: (**a**) slender elements from series AS1C1 and AS1C2 (different concrete strength), (**b**) moderately slender columns made from the same grade of concrete and different grades of steel, series DS1 and DS2.

**Figure 17 materials-15-02231-f017:**
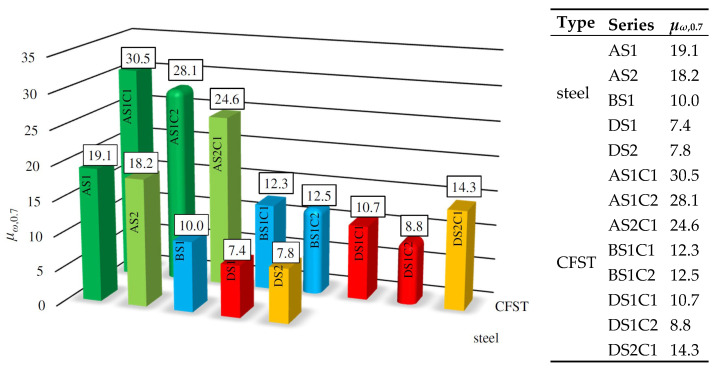
Ductility of steel and CFST columns calculated using proposed *μ_ω,_*_0.7_ measure.

**Figure 18 materials-15-02231-f018:**
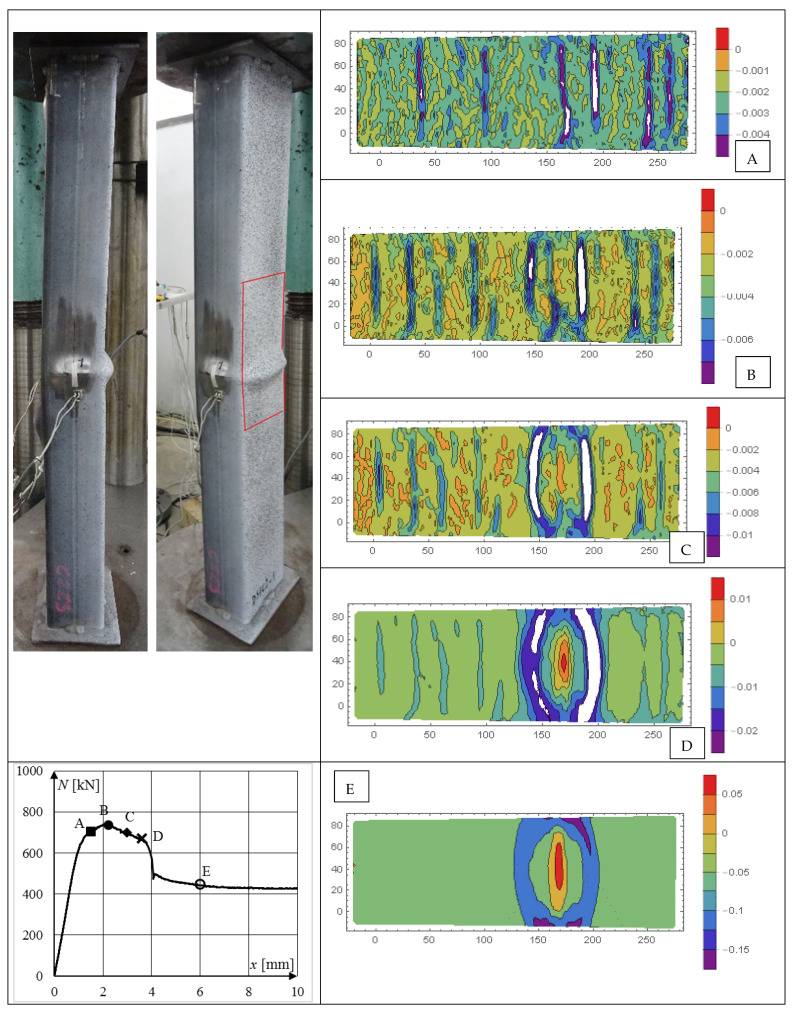
Evolution of strains *ε_xx_* in column DS1C2_1, steel S235, stronger concrete, *ξ* = 0.90.Points (**A**,**B**) indicate strains before reaching the ultimate load and when the maximum force was reached, respectively. Points (**C**–**E**) mark selected places in the post-peak part of the equilibrium path.

**Figure 19 materials-15-02231-f019:**
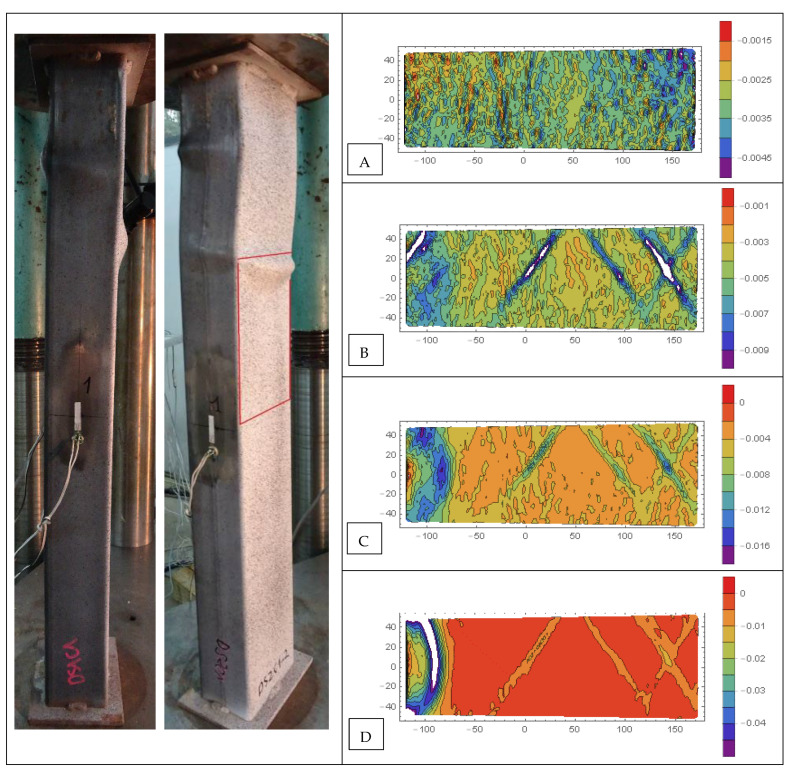
Evolution of strains *ε_xx_* in column DS2C1_2, steel S355, weaker concrete, *ξ* = 1.51. Points (**A**,**B**) indicate strains before reaching the ultimate load and when the maximum force was reached, respectively. Points (**C**–**E**) mark selected places in the post-peak part of the equilibrium path.

**Figure 20 materials-15-02231-f020:**
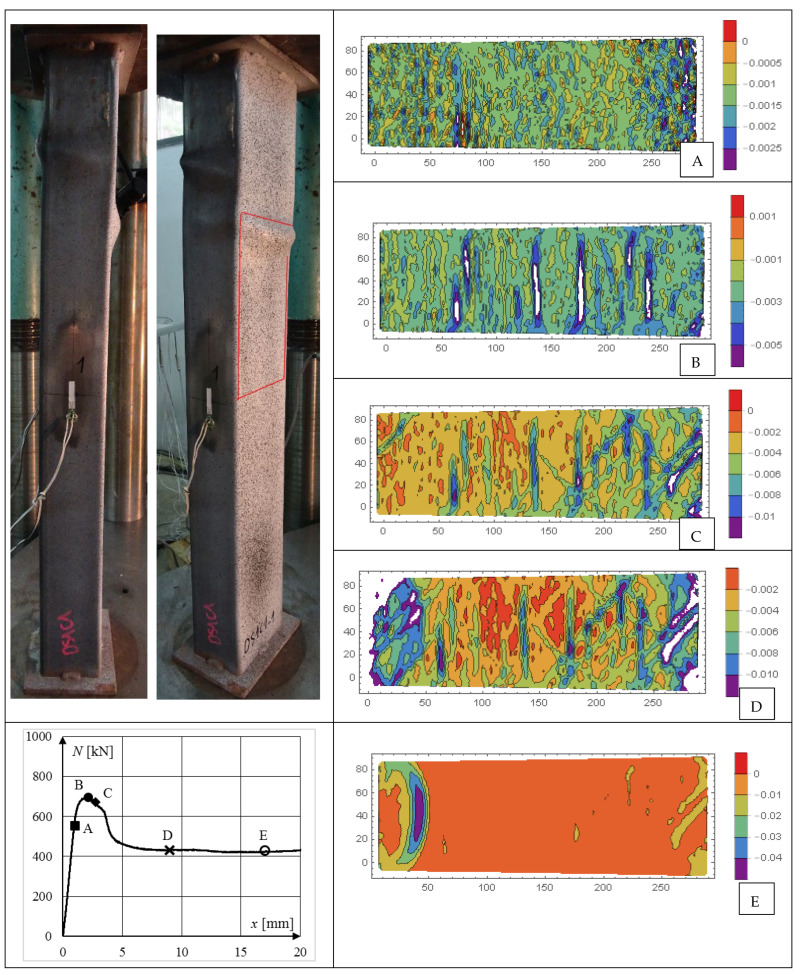
Evolution of strains *ε_xx_* in column DS1C1_1, steel S235, weaker concrete, *ξ* = 1.16. Points (**A**,**B**) indicate strains before reaching the ultimate load and when the maximum force was reached, respectively. Points (**C**–**E**) mark selected places in the post-peak part of the equilibrium path.

**Figure 21 materials-15-02231-f021:**
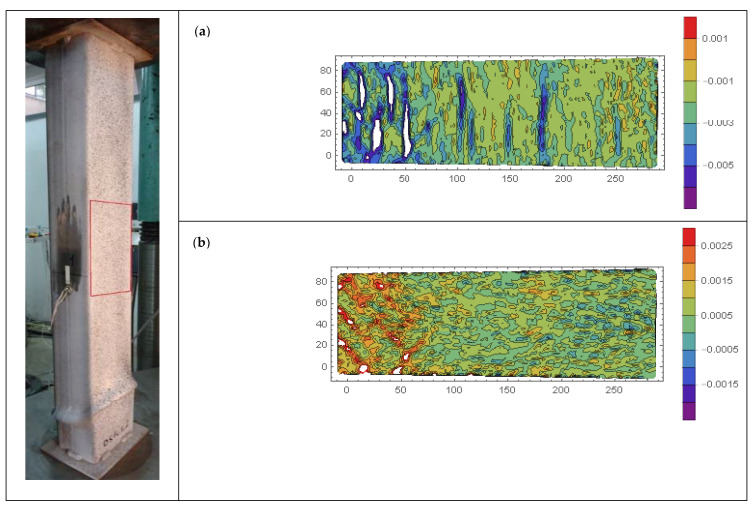
(**a**) Longitudinal *ε_xx_* and (**b**) transverse *ε_yy_* strains in column DS1C1_2 captured in the same point of the equilibrium path.

**Figure 22 materials-15-02231-f022:**
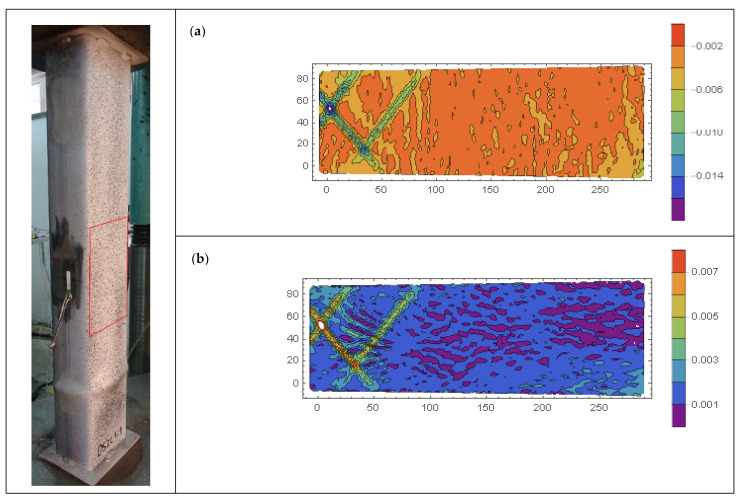
(**a**) Longitudinal *ε_xx_* and (**b**) transverse *ε_yy_* strains in column DS2C1_1 captured in the same point of the equilibrium path.

**Figure 23 materials-15-02231-f023:**
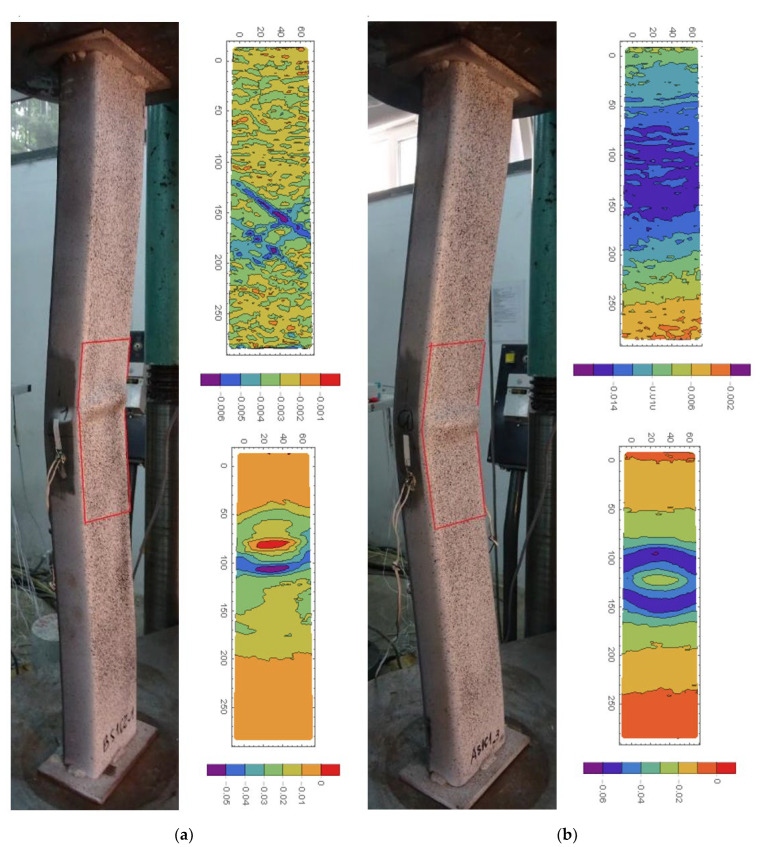
Longitudinal strains *ε_xx_* in slender columns: (**a**) BS1C2_1 (*ξ* = 1.52) and (**b**) AS1C1_3 (*ξ* = 4.12). Description in the text.

**Table 1 materials-15-02231-t001:** Material and geometric properties of the investigated columns—experimental research.

Series	Steel Cross-Section	Steel Grade	Yield Strength *f_y_* (MPa)	Concrete Strength *f_c_* (MPa)	*D*/*t*/*L*/*D*	$\bar{\lambda }$	*ξ*	Quantity
DS1	RHS 120 × 80 × 3	S235	317.0	-	26.7/9.4	0.29	-	1
DS1C1		S235	317.0	37.0		0.32	1.16	3
DS1C2		S235	317.0	47.9		0.34	0.90	3
DS2		S355	414.0	-		0.33	-	1
DS2C1		S355	414.0	37.0		0.35	1.51	2
AS1	RHS 100 × 50 × 5	S235	408.3	-	10.0/15	0.54	-	1
AS1C1		S235	408.3	37.0		0.56	4.12	3
AS1C2		S235	408.3	47.9		0.57	3.18	3
AS2		S355	426.2	-		0.55	-	1
AS2C1		S355	426.2	37.0		0.57	4.30	2
BS1	RHS 100 × 50 × 3	S235	357.4	-	16.7/15	0.48	-	1
BS1C1		S235	357.4	37.0		0.53	1.97	3
BS1C2		S235	357.4	47.9		0.54	1.52	2

## Data Availability

The data presented in this study are available on request from the corresponding author.

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
