# Peer review of "Experimental Investigation on the Vertical Ductility of Rectangular CFST Columns Loaded Axially"

_materials, 2022, doi:10.3390/ma15062231_

Round 1

Reviewer 1 Report

The reviewer accept the present form beside the error of reference citation in the paper.

This paper contributes the investigation of vertical ductility of CFST composite column. The authors provided a couple of comments that should be addressed before publishing:

1.     It is appropriate to explain the full name of “CFST” at the beginning of the manuscript, such as Abstract.

2.     The whole manuscript needs some polishing works.

3.     The current manuscript includes so much “Error! Reference source not found” and this problem must be solved before publishing.

4.     Line 91 it has section title “1. Ductility of CFST columns – state of the art”, while line 232 has the title “1. Proposal ..”. Both of them have the section number 1 which are really confused. Need re-manage the section numbers in the entire manuscript.

5.     Line 550 Is it Figure 1? Please pay enough attention on the figure numbers and ensure the cited figure is correct.

6.     Conclusion sections are too poor to be published. Please rewrite the entire section with a clear and concise language.

Author Response

Dear Reviewer,

The authors wish to thank You for all detailed comments on our work. All the remarks have been addressed in the revised manuscript or in the detailed response to Reviewers’ comments, as indicated below. The changes in text are marked in blue in the manuscript.

Best Regards,

Bartosz Grzeszykowski, Elżbieta Szmigiera

Reviewer 2 Report

Thank you for this contribution. This is an interesting and timely manuscript. This paper presents results from tests on CFST columns. The conducted tests and analysis are typically standard and fall within the expected work from such a publication and hence the work merits publication. As such, the authors are invited to properly address the following items:

  1. In general, the introduction is light and does not represent the state of the art in this domain. The amount of work in this area continues to rapidly rise. The authors are advised to strengthen their literature review section with supplementary material. For example, consider the works by the Zarringol et al. group on CFST testing and modeling, among others.
  2. How were the digital image correlation calibrated in the tests?
  3. What would the boundary conditions of the tested columns be classified under?
  4. The conclusions section is mistakenly numbered as 1.0.
  5. Please fix the " Error! Reference source not found." errors throughout the whole paper.

Author Response

(The authors gave the same response as above.)

Reviewer 3 Report

This is an interesting paper, however, its presentation style is poor and hard to follow. I personally find many mistakes and ambigious descriptions, and I suggest the authors carefully revise this paper. The followings are some points they need to address.

  1. In line 62-63, the sentence “everal definitions of vertical ductility measures of columns have been used in the literature. (see Error! Reference source not found.)” requires to be revised by placing the reference the authors want to cite. The similar things happen in lines 72-74, lines 92-93, lines 118-119, lines 157-159 and many other paragraphs. Please check them all and correct them.
  2. When the abbreviation “CFST” is used, the full name should be given.
  3. In Fig.8, it seems that Ew is the area under the curve from x=x0 to x=xw, which is not the same as the formula. Please revise the figure.
  4. The authors mentioned the columns are moderate slenderness. Please give a clear definition of moderate slenderness.
  5. The mixtures of concrete cores should be given. In addition, the bond strengths between concretes and steel should be prescribed. The compressive strength of concretes should be addressed.
  6. The lables for figures in section 1.1 are wrong. Please check it.

Author Response

(The authors gave the same response as above.)

Round 2

Reviewer 2 Report

Thank you for revising your work.

Reviewer 3 Report

The authors have revised the manuscript carefully and respond to all the points I mentioned in the 1st round report. It is now my pleasure to recommend its acceptance.